# Adversarial Teacher-Student Representation Learning for Domain Generalization

**Fu-En Yang**[1,2]     **Yuan-Chia Cheng**[1]     **Zu-Yun Shiau**[1]     **Yu-Chiang Frank Wang**[1,2]

[1]Graduate Institute of Communication Engineering, National Taiwan University, Taiwan
[2]ASUS Intelligent Cloud Services, Taiwan
{f07942077, r08942154, r09942069, ycwang}@ntu.edu.tw

## Abstract

Domain generalization (DG) aims to transfer the learning task from a single or multiple source domains to unseen target domains. To extract and leverage the information which exhibits sufficient generalization ability, we propose a simple yet effective approach of Adversarial Teacher-Student Representation Learning, with the goal of deriving the domain generalizable representations via generating and exploring out-of-source data distributions. Our proposed framework advances Teacher-Student learning in an adversarial learning manner, which alternates between knowledge-distillation based representation learning and novel-domain data augmentation. The former progressively updates the teacher network for deriving domain-generalizable representations, while the latter synthesizes data out-of-source yet plausible distributions. Extensive image classification experiments on benchmark datasets in multiple and single source DG settings confirm that, our model exhibits sufficient generalization ability and performs favorably against state-of-the-art DG methods.

## 1 Introduction

Deep neural networks have achieved promising performance on a wide variety of tasks. However, these networks assume the training and testing samples fall in the same data distribution. Such a strong assumption would limit the applicability of the learned models in real-world scenarios (e.g., cross-city autonomous driving or multi-cite medical imaging task), in which training and testing data are typically observed under different conditions. In other words, the generalizability of the model at *unseen* target domains might be poor due to *unexpected domain shifts*. To tackle the domain discrepancy, domain generalization (DG) has been proposed and drawn increasing attention recently.

The aim of DG is to train models using data observed from single or multiple source domains, while expecting that the model would be generalized to unseen target domains. Most existing DG approaches focus on deriving domain-invariant features [1] among multiple source domains or adopting meta-learning techniques [2, 3, 4, 5], which would simulate domain shifts during the meta-training stage. However, the features derived by the above methods are generally guaranteed to be invariant to the seen source domains, not the generalizability of the learned representation to describe unseen domain data. To overcome the limitation, [6, 7, 8, 9] turn to leverage data generation techniques for diversifying the source distributions, and thus avoid overfitting on source domains yet improve the generalization ability of models. Specifically, several works [6, 7, 8] choose to generate novel-domain images by either perturbing the style of source data to confuse the domain classifier [6, 7], or transporting the source data to novel styles via optimal transport based objective [8]. [9] adopts Mixup [10] to interpolate the feature statistics between samples from different domains. While the above methods perform well, designing an objective for generating samples with DG guarantees remains a challenging and open problem.

35th Conference on Neural Information Processing Systems (NeurIPS 2021).

Recently, self-supervised pre-training manifests the potential to derive generalizable representation, which serves as a promising start point for downstream tasks (e.g., image segmentation or object detection). In domain generalization, a number of self-supervision techniques have been introduced [11, 12, 13] to improve network transferability by discovering the intrinsic properties within images. For instance, [11, 12, 13] adopt jigsaw puzzles as the pretext task, which predicts the relative positions of image patches to constrain the semantic feature learning in a multi-task training fashion. Recently, contrastive learning approaches [14, 15, 16, 17, 18] have been proposed and widely applied, which establish the representation learning from multiple views of an image to extract the task-relevant information and discard task-irrelevant noise. However, the concept of such multiview learning [19, 18] is simply realized by hand-crafted image transformations (e.g., *RandomResizedCrop, Color Jittering, or Gaussian Blur*). The effectiveness of these hand-crafted image transformations for benefiting the generalization to unseen distributions is still not guaranteed.

In this paper, we propose a unique *Adversarial Teacher-Student Representation Learning* framework for tackling domain generalized visual classification. Based on the recent success of contrastive learning, we advance the concept of multi-view learning into DG regime for augmenting source instances to out-of-source styles and diversifying training distributions. To be more precise, with the goal of learning representations which are robust to unseen domain shift, we propose to jointly perform *Domain Generalized Representation Learning* and *Novel Domain Augmentation* in an adversarial learning manner. Based on Teacher-Student learning schemes [20, 16, 21], our framework utilizes original images as inputs to the teacher network and takes stylized augmentations as input to the student network. To ensure both learning stages produce domain generalized representation, we adopt the Teacher-Student co-training scheme, which progressively refines Teacher by the distilled knowledge learned from Student by observing augmented novel-domain data, enabling Teacher to be generalizable to data with out-of-source distributions. On the other hand, *Adversarial Novel Domain Augmentation* aims at augmenting unseen domain data using source-domain training instances. The objective is to *maximize* the discrepancy between the input and augmented data, derived from the teacher and student modules, respectively. In order to have such augmented data exhibit sufficient domain differences, the above discrepancy will be calculated using features derived from data across different source domains. By iteratively training the above two stages in an adversarial learning fashion, the resulting model (Teacher) would be able to derive domain generalizable representations.

The contributions of this paper are highlighted as below:

- Different from existing meta-learning based approaches, we propose a teacher-student adversarial learning scheme for addressing domain generalization classification problems.

- In the stage of Domain Generalized Representation Learning, the student network observes augmented novel-domain data and distills the information to update the teacher network, allowing derivation of domain generalizable representation.

- In the stage of Novel Domain Augmentation, the generator aims at producing unseen yet plausible domain data, which maximizes the discrepancy between augmented and existing domains while the semantic information is preserved.

- Evaluations on several benchmark datasets in multiple and single source domain settings verify that our method performs favorably against existing DG approaches and exhibits sufficient domain generalization capability.

## 2 Related Works

**Domain Generalization (DG)**    Different from domain adaptation (DA), which observes both source and target-domain training data for performing learning tasks across domains [22, 23, 24, 25, 26], DG deals with a more realistic yet challenging setting. More precisely, DG aims at generalizing the model trained only on single or multiple source domains to recognize the test instance in unseen but similar target domain. With only source-domain data observed during training, a number of works [3, 2, 27] apply meta-learning for learning domain-invariant features. These methods typically partition source domains into meta-train and meta-test splits to simulate the domain shifts during training. Feature-Critic [27] meta-learns a critic network to evaluate the generalized degree of extracted features for encouraging robust feature derivation. [4] introduces an episodic training that cross-trains domain-specific feature extractors and classifiers to let the learned model invariant to the domain shift. MLDG [2] and MASF [5] both adopt gradient based meta-learning to simulate

the domain shift, while [5] additionally enforces local and global constraints in meta-training. In addition to meta-learning approaches, [11] jointly solves jigsaw puzzle as an auxiliary task with standard classification in a multi-task fashion. RSC [28] iteratively discards the dominant features on the training data to improve generalization. Nevertheless, these approaches employ solely limited source domains to derive generalizable features, which still draws a concern about over-fitting on source domains [8, 29] and restricts the generalization ability to unseen domains.

Recent research works consider data generation as an alternative technique for domain generalization, which increase the diversity of training data distribution. To achieve this goal, [6, 7] are inspired by adversarial attack [30]. CrossGrad [6] perturbs source data by adding adversarial gradients; DDAIG [7] learns a transformation network that outputs perturbations to confuse the domain classifier. However, such perturbed images do not necessarily exhibit sufficient data domain diversity. In contrast of adding perturbation to images, L2A-OT [8] learns a conditional generator that transforms images from a source distribution to a pseudo-novel distribution by an optimal transport based objective. MixStyle [9] produces image features with mixed feature statistics across source domains. Very recently, PDEN [31] utilizes a progressive learning strategy for single-source domain generalization, which iteratively expands the training data set by adding augmented data. Note that although they adopt contrastive and adversarial learning objectives which are similar to ours, our proposed approach is able to tackle both multi-source and single-source DG problems, and also comes with superior memory efficient performance and comparably stable training process.

**Self-Supervised Learning (SSL)**    Self-supervision is a recent paradigm for unsupervised learning. The idea is to design pretext tasks for feature learning to facilitate the downstream task learning. Such auxiliary pretext tasks can be predictions of the image colors [32], relative locations of patches from the same image [11, 12], and image rotation [33]. Very recently, contrastive learning [14, 15, 16, 17, 18] has achieved promising results on network pre-training to learn generalized image features. [19, 18] reveal that the success of contrastive learning is typically built on the multi-view perspective, and prove theoretically and experimentally that the compact and robust representations can be learned by deriving the invariance among multiple views of an image. We adapt the above concept of multi-view learning into DG regime. We focus on learning novel-domain data augmentations across source-domain instances in an adversarial training fashion. As detailed and verified later, our proposed learning scheme would produce domain generalizable representation for unseen target-domain data, and performs favorably against state-of-the-art DG approaches.

## 3   Proposed Method

### 3.1   Problem Formulation and Model Overview

For the sake of completeness, we first define the problem setting and notations used in this paper. We assume that training data are observed from $N$ source domains $\mathcal{D}_{tr} = \{\mathcal{D}_1, \mathcal{D}_2, ..., \mathcal{D}_N\}$, each of which contains a set of image and label pairs $\mathcal{D}_i = \{X_i, Y_i\}$. Our goal is to learn a model which would exhibit sufficient generalization capability, so that classification of test data in unseen target domains can be performed. In order to derive domain-generalized feature representations, we present a novel *Adversarial Teacher-Student Representation Learning* framework, which is a min-max deep learning framework alternating between the following two stages: *domain generalized representation learning* (Sec. 3.2) and *novel domain augmentation* (Sec. 3.3), as depicted in Fig. 1. For *domain generalized representation learning*, we learn a domain-generalized teacher network (Teacher) $F_T$ with the help from a student network (Student) $F_S$, which observes synthesized *novel*-domain augmentation and distills knowledge to Teacher. As for *novel-domain augmentation*, the novel-domain augmenter $G$ is learned to observe the discrepancy of Teacher-Student encoders, which progressively generates *strong* novel stylized augmentations to diversify training distributions. Once the learning of the above framework is complete, the teacher network would extract domain-generalized features for the task network (e.g., classifier), and thus classification of unseen target-domain data can be performed accordingly. We now detail our proposed learning schemes in the following subsections.

### 3.2   Teacher-Student Domain Generalized Representation Learning

While techniques based on learning across multiple source domains for DG exist (e.g., using meta-learning techniques like [3, 4, 5]), it is not clear how the learned model and feature representations

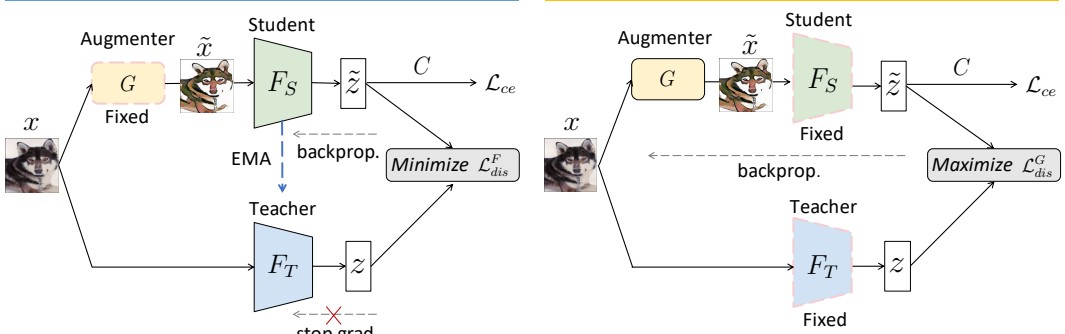

Figure 1: Overview of our Adversarial Teacher-Student Representation Learning scheme, which includes the teacher network $F_T$, the student network $F_S$, classifier $C$, and novel-domain augmenter $G$. Note that we alternate between the stages of domain generalized representation learning and novel-domain augmentation in a mutually beneficial manner, resulting in discriminative yet domain generalized representations.

would be generalized to unseen target domains. Instead of directly fitting models across source domains, we propose *Domain Generalized Representation Learning* based on the Teacher-Student learning scheme, with the goal of extracting domain generalizable feature representations. To ensure our teacher encoder to gain generalizability by observing out-of-source domain information, we deploy a Student $F_S$ for exploring novel-domain augmentation synthesized from the novel-domain augmenter $G$, while distilling the associated knowledge to update $F_T$.

To address this representation learning task, we first train the teacher module together with a single-layer classifier $C$ using multiple source-domain data. The standard cross-entropy loss $\mathcal{L}_{ce}$ is utilized to initialize $F_T$ as warm-up. As illustrated in Fig. 1, we then input training images $x$ sampled from the source domains into the novel-domain augmenter $G$ (detailed in the following sub-section), producing the style (or domain) perturbed augmentation $\tilde{x}$ yet preserving its semantic information. While such a domain augmented $\tilde{x}$ would be fed into the student module resulting in feature $\tilde{z} = F_S(\tilde{x})$, we also feed the original input $x$ into Teacher to derive $z = F_T(x)$. To ensure that $\tilde{z}$ would contain the same semantic information as $z$ does, we particularly propose an objective to **minimize** the discrepancy between $\tilde{z}$ and $z$. To be more specific, we define the discrepancy loss $\mathcal{L}_{dis}^F$ to minimize the distance between the normalized features $\tilde{z}$ and $z$:

$$\min_{F_S} \mathcal{L}_{dis}^F(z, \tilde{z}) = \left\| \frac{z}{\|z\|_2} - \frac{\tilde{z}}{\|\tilde{z}\|_2} \right\|_2^2 = \left\| \frac{F_T(x)}{\|F_T(x))\|_2} - \frac{F_S(\tilde{x})}{\|F_S(\tilde{x})\|_2} \right\|_2^2. \tag{1}$$

In addition, we calculate the cross-entropy loss on the domain-augmented feature $\tilde{z}$, i.e., $\mathcal{L}_{ce}(C(\tilde{z}), y)$, which further enforces the classification capability of the student module (note that $C$ indicates the single-layer classifier, and $y$ denotes the corresponding class label). We note that, in this representation learning stage, only the student network $F_S$ is updated by the above two objectives $\mathcal{L}_{dis}^F(z, \tilde{z})$ and $\mathcal{L}_{ce}(C(\tilde{z}), y)$, and we apply a stop-gradient strategy to forbid $F_T$ and $G$ from being updated by gradients. Thus, optimization of $F_S$ with learning rate $\gamma$ can be expressed as follows:

$$\theta_S \leftarrow \theta_S - \gamma \frac{\partial(\mathcal{L}_{dis}^F(z, \tilde{z}) + \mathcal{L}_{ce}(C(\tilde{z}), y))}{\partial \theta_S}. \tag{2}$$

As for the teacher network $F_T$, we adopt exponential moving average (EMA) [20, 16, 21] to progressively refine the associated model parameter $\theta_T$. That is, the learned knowledge from Student's parameter $\theta_S$ is distilled to update $\theta_T$ as follows,

$$\theta_T \leftarrow \tau \theta_T + (1 - \tau)\theta_S, \quad \text{where } \tau \in [0, 1), \tag{3}$$

Note that $\tau$ controls the updates on the teacher network. Finally, it is also worth pointing out that, such a refinement strategy would avoid the teacher module from directly observing unrealistic domain augmentations, which might degrade its domain generalization capability.

## 3.3 Adversarial Novel Domain Augmentation

To motivate the student network to explore sufficient diversity of domain augmentation, we present an adversarial learning scheme, which would progressively perform novel-domain data augmentation in our proposed framework. Inspired by both adversarial learning strategy [34] and multiview learning from SSL [19, 18], we formulate our novel-domain augmentation stage together with representation learning (Sec. 3.2) into an adversarial learning framework. As depicted in the right hand side of Fig. 1, we aim at training the novel-domain augmenter $G$ and freezing both $F_T$ and $F_S$, while the discrepancy between $z$ and $\tilde{z}$ serves as the adversarial guidance. That is, when the above discrepancy is small (i.e., the outputs of Teacher and Student are similar), it implies that such domain augmentations have been seen by existing source-domain data. To encourage more the augmented data to be sufficiently distinct in terms of domain information, we train our novel-domain augmenter by **maximizing** the discrepancy as follows,

$$\max_G \mathcal{L}_{dis}^G(z, \tilde{z}) = [\left\| \frac{z}{\|z\|_2} - \frac{\tilde{z}}{\|\tilde{z}\|_2} \right\|_2^2 - m]_- = [\left\| \frac{F_T(x)}{\|F_T(x)\|_2} - \frac{F_S(G(x))}{\|F_S(G(x))\|_2} \right\|_2^2 - m]_-, \quad (4)$$

where $[\cdot]_- = min(\cdot, 0)$, and the margin $m$ can either be calculated by the means/centroids of data from each source domain in a mini-match, followed by averaging the L2 distances between the above centroid pairs, or simply viewed as a hyperparameter. It is worth pointing out that, this margin serves as a regularization observed from the separation between existing source domains. Thus, it reflects the desirable domain gap between the augmented and existing domain data.

To guarantee the produced domain augmentations preserve the original categorical content, we still observe the cross-entropy loss $\mathcal{L}_{ce}(C(\tilde{z}), y)$ with regard to $C(\tilde{z})$ and the corresponding label $y$. Thus, optimization of $G$ can be performed as follows,

$$\theta_g \leftarrow \theta_g - \gamma \frac{\partial(-\mathcal{L}_{dis}^G(z, \tilde{z}) + \mathcal{L}_{ce}(C(\tilde{z}), y))}{\partial \theta_g}. \quad (5)$$

Note that, we only pretrain the classifier using source domain data available, and keep it fixed during the learning of our teacher-student augmentation framework. If we allow the update of this classifier during the training process, it might observe undesirable outputs and affect the learning of both augmenter and teacher/student modules in the early training stage, where either the augmented data or its extracted features is not yet quality.

Once the learning of the proposed framework is complete (i.e., alternative optimization between the two stages), we deploy the derived domain generalized Teacher to extract discriminative and transferable features, so that classification of unseen target domain can be performed accordingly. The pseudo code of our Adversarial Teacher-Student Representation Learning is summarized in Algorithm 1 in supplementary material.

## 4 Experiments

### 4.1 Datasets and Evaluation Protocol

**Datasets**   We evaluate our method on several public benchmark datasets. **PACS** [35] is composed of four data domains, *Photo*, *Art painting*, *Cartoon* and *Sketch*, with diverse image colors and styles. Each domain contains 7 categories, with 9991 images in total. Following the experimental protocol proposed by [35], images from source domains are divided into the training split and the validation split, at a ratio of about 9:1. **Office-Home** [36] is comprised of four domains, *Art*, *Clipart*, *Product* and *Real world*, and exists larger label sets of 65 categories, with about 15500 images in total. The dataset contains images of everyday objects with various styles, backgrounds and camera viewpoints. Images are divided into the training split and the validation split at a ratio of about 9:1. **DomainNet** [37] is a recently proposed large-scale dataset which consists of 0.6 million images of 345 classes distributed across 6 domains, *Real*, *Clipart*, *Infograph*, *Painting*, *Quickdraw* and *Sketch*. We follow the training and testing splits for all the 6 domains released by [37]. Also, for the single source DG experiments, we follow [38] and partition the training split from [37] into the training and validation splits at a ratio of 9:1. Due to page limitation, we additionally provide quantitative comparisons on **VLCS** [39] and **Digit-DG** [7] datasets in the supplementary material.

Table 1: Comparisons to non-data-generation based methods on PACS using ResNet-18 in leave-one-domain-out settings. **Bold** denotes the best result.

| Target | DeepAll (baseline) | MMD-AAE [1] | MLDG [2] | JiGen [11] | MetaReg [3] | Epi-FCR [4] | MASF [5] | EISNet [12] | DMG [38] | Borlino et al. [44] | DSON [45] | RSC [28] | Ours |
|---|---|---|---|---|---|---|---|---|---|---|---|---|---|
| Photo | 95.6 | 96.0 | 96.1 | 96.0 | 95.5 | 93.9 | 95.0 | 95.9 | 93.4 | 95.0 | 95.9 | 96.0 | **97.3** $\pm$ 0.3 |
| Art painting | 75.1 | 75.2 | 81.3 | 79.4 | 83.7 | 82.1 | 80.3 | 81.9 | 76.9 | 82.7 | 84.7 | 83.4 | **85.8** $\pm$ 0.6 |
| Cartoon | 74.2 | 72.7 | 77.2 | 75.3 | 77.2 | 77.0 | 77.2 | 76.4 | 80.4 | 78.0 | 77.7 | 80.3 | **80.7** $\pm$ 0.5 |
| Sketch | 68.4 | 64.2 | 72.3 | 71.4 | 70.3 | 73.0 | 71.7 | 74.3 | 75.2 | 81.6 | **82.2** | 80.9 | 77.3 $\pm$ 0.5 |
| Average | 78.3 | 77.0 | 81.8 | 80.5 | 81.7 | 81.5 | 81.1 | 82.2 | 81.5 | 84.3 | 85.1 | 85.2 | **85.3** |

Table 2: Comparisons to non-data-generation based methods on Office-Home using ResNet-18 in leave-one-domain-out settings. **Bold** denotes the best result.

| Target | DeepAll (baseline) | CCSA [46] | MMD-AAE [1] | MLDG [2] | D-SAM [47] | JiGen [11] | Borlino et al. [44] | DSON [45] | RSC [28] | Ours |
|---|---|---|---|---|---|---|---|---|---|---|
| Art | 59.0 | 59.9 | 56.5 | 58.1 | 58.0 | 53.0 | 58.7 | 59.4 | 58.4 | **60.7** $\pm$ 0.5 |
| Clipart | 48.4 | 49.9 | 47.3 | 49.3 | 44.4 | 47.5 | 52.3 | 45.7 | 47.9 | **52.9** $\pm$ 0.3 |
| Product | 72.5 | 74.1 | 72.1 | 72.9 | 69.2 | 71.5 | 73.0 | 71.8 | 71.6 | **75.8** $\pm$ 0.1 |
| Real world | 75.5 | 75.7 | 74.8 | 74.7 | 71.5 | 72.8 | 75.0 | 74.7 | 74.5 | **77.2** $\pm$ 0.2 |
| Average | 63.9 | 64.9 | 62.7 | 63.8 | 60.8 | 61.2 | 64.8 | 62.9 | 63.1 | **66.7** |

**Evaluation Protocol** For fair comparison purposes, we follow the leave-one-domain-out protocol as considered in [7, 8, 12, 9] for our experiments. That is, one data domain from a dataset is selected as the target unseen domain to be recognized, and the remaining ones as the source domains for training. And, we report the top-1 classification accuracy (%) for quantitative evaluation.

## 4.2 Implementation Details

For PACS, Office-Home, and DomainNet, input images are resized to $224 \times 224$ pixels, and we use ResNet-18 and ResNet-50 [40] pre-trained on ImageNet [41] as the backbones of our teacher and student networks. $F_S$ is trained with the SGD optimizer, with an initial learning rate of 0.0005, and a batch size of 32 for 60 epochs. The learning rate is decayed by 0.1 after 30 epochs. $F_T$ is updated via EMA with the momentum coefficient $\tau$ of 0.999 by default. Our novel-domain augmenter $G$ is realized by a fully convolutional network similar to the generator's architecture in [7] and trained with the SGD optimizer. In the warm-up phase, we train $F_T$ together with the classifier $C$ using only source data with the SGD optimizer, and then the parameters of $C$ are fixed in the following training process. Note that we also use the official implementation from [11, 6, 7, 42, 8, 9] for our comparisons. In all our experiments, we implement our model using PyTorch and Dassl.pytorch [43] toolbox, and conduct training on a single NVIDIA TESLA V100 GPU with 32 GB memory.

## 4.3 Quantitative Evaluation

We first perform domain-generalized visual classification tasks and compare our results with existing *non-data-generation* [1, 46, 2, 11, 3, 4, 5, 12, 38, 44, 45, 28] and *data-generation* based [6, 7, 8, 9] methods on two commonly-used public benchmarks, **PACS** and **Office-Home**. In our experiments, *DeepAll* is viewed as a baseline, in which both feature extractor and classifier are trained on data aggregated from all source domains.

Tables 1 and 2 summarize the quantitative comparisons with existing *non-data-generation* based methods [1, 46, 2, 11, 3, 4, 5, 12, 38, 44, 45, 28] on PACS and Office-Home (ResNet-18 as the backbone), respectively. Particularly, Epi-FCR [4] and MASF [5] are meta-learning approaches which either adopt episodic training scheme that cross-train encoders and classifiers from different domains, or employ a gradient-based optimization strategy with global and local losses for regularizing the model training. JiGen [11] and EISNet [12] both consider solving jigsaw puzzles as the auxiliary task for better capturing spatial information. Recent start-of-the-art method RSC [28] iteratively dropouts the most contributing features to force models to explore the remaining features that correlate with semantic information. As we can observe from Table 1, our approach achieved the best performance on *Photo*, *Art paining* and *Cartoon*. It is worth noting that, a significant gap in visual appearance can be seen between *Sketch* and other image domains, which makes the associated domain generalization more difficult. Nevertheless, our approach still achieved satisfactory results over the state-of-the-art methods on *Sketch*, and reported the highest average accuracy of **85.3%**. On the other hand, Table 2 demonstrates that our method performed favorably on all the domains (i.e., 60.7% on *Art*, 52.9% on *Clipart*, 75.8% on *Product*, and 77.2% on *Real world*), and thus achieves the highest average

Table 3: Comparisons to data-generation based methods on PACS using ResNet in leave-one-domain-out settings. **Bold** denotes the best result.

| Target | ResNet-18 | | | | | | ResNet-50 | | | | |
|---|---|---|---|---|---|---|---|---|---|---|---|
| | DeepAll (baseline) | CrossGrad [6] | DDAIG [7] | L2A-OT [8] | MixStyle [9] | Ours | DeepAll (baseline) | CrossGrad [6] | DDAIG [7] | MixStyle [9] | Ours |
| Photo | 95.6 | 96.0 | 95.3 | 96.2 | 96.1 | **97.3** $\pm$ 0.3 | 94.8 | 97.8 | 95.7 | 98.0 | **98.9** $\pm$ 0.3 |
| Art painting | 75.1 | 79.8 | 84.2 | 83.3 | 84.1 | **85.8** $\pm$ 0.6 | 81.5 | 87.5 | 85.4 | 87.4 | **90.0** $\pm$ 0.3 |
| Cartoon | 74.2 | 76.8 | 78.1 | 78.2 | 78.8 | **80.7** $\pm$ 0.5 | 78.6 | 80.7 | 78.5 | 83.3 | **83.5** $\pm$ 0.5 |
| Sketch | 68.4 | 70.2 | 74.7 | 73.6 | 75.9 | **77.3** $\pm$ 0.5 | 69.7 | 73.9 | **80.0** | 78.5 | **80.0** $\pm$ 0.6 |
| Average | 78.3 | 80.7 | 83.1 | 82.8 | 83.7 | **85.3** | 81.2 | 85.7 | 84.9 | 86.8 | **88.1** |

Table 4: Comparisons to data-generation based methods on Office-Home using ResNet in leave-one-domain-out settings. **Bold** denotes the best result.

| Target | ResNet-18 | | | | | | ResNet-50 | | | | |
|---|---|---|---|---|---|---|---|---|---|---|---|
| | DeepAll (baseline) | CrossGrad [6] | DDAIG [7] | L2A-OT [8] | MixStyle [9] | Ours | DeepAll (baseline) | CrossGrad [6] | DDAIG [7] | MixStyle [9] | Ours |
| Art | 59.0 | 58.4 | 59.2 | 60.6 | 58.7 | **60.7** $\pm$ 0.5 | 64.7 | 67.7 | 65.2 | 64.9 | **69.3** $\pm$ 0.2 |
| Clipart | 48.4 | 49.4 | 52.3 | 50.1 | **53.4** | 52.9 $\pm$ 0.3 | 58.8 | 57.7 | 59.2 | 58.8 | **60.1** $\pm$ 0.6 |
| Product | 72.5 | 73.9 | 74.6 | 74.8 | 74.2 | **75.8** $\pm$ 0.1 | 77.9 | 79.1 | 77.7 | 78.3 | **81.5** $\pm$ 0.4 |
| Real world | 75.5 | 75.8 | 76.0 | 73.0 | 75.9 | **77.2** $\pm$ 0.2 | 79.0 | 80.4 | 76.7 | 78.7 | **82.1** $\pm$ 0.2 |
| Average | 63.9 | 64.4 | 65.5 | 65.6 | 65.5 | **66.7** | 70.1 | 71.2 | 69.7 | 70.2 | **73.3** |

accuracy **66.7%**. The above quantitative comparisons verify that, comparing to directly (meta-)learn from existing source domain data, our approach for augmenting diverse, novel, yet semantically practical source-domain training data would be preferable in domain generalization tasks.

With the above observation, we further compare our method with the state-of-the-art *data-generation* based models [6, 7, 8, 9] using ResNet-18 and ResNet-50 as backbones. As shown in Table 3, our approach consistently performed superiorly against the method of [9] by 1.6% and 1.3% on PACS with ResNet-18 and ResNet-50 backbones, respectively. Table 4 presents the results on Office-Home, which shows that our method would be preferable among the DG methods considered. Also, the above results demonstrate that our proposed framework is able to achieve general preferable performances regardless of the backbone choice. It is worth noting that, CrossGrad [6] and DDAIG [7] add perturbation to input images, which might not represent the domain variations, and the data generation processes of L2A-OT [8] do *not* jointly take the representation learning into consideration. Also, MixStyle [9] can only produce image features with *interpolated* domain styles. Different from these methods, our approach learns to synthesize out-of-source distribution augmentations and derive domain generalized representations in a mutually beneficial manner, hence exhibiting more robust generalization capability.

## 4.4 Analysis of Our Method

### 4.4.1 Ablation Study

We now conduct the ablation study to verify our network design on PACS with ResNet-50 backbone, and we list the results in Table 5. Also, we evaluate the effectiveness of *Jigsaw puzzle*. Such spatial transformation has been applied in several DG works [11, 12]. In the bottom part of Table 5, we consider different network designs, including *Siamese architecture*, *Student without EMA*, and *Student with EMA*, to be derived for performing on unseen target domains.

**Effectiveness of Adversarial Augmenter**   In the upper part of this table, we first verify the effectiveness of our designed *novel-domain augmenter $G$* by replacing $G$ with different types of data augmentation strategies *Random Augmentation* and *Jigsaw puzzle*. *Random Augmentation* denotes directly performing hand-crafted image transformations, including RandomResizedCrop, Color Jittering, Gaussian Blur, RandAugment, and Color Dropping. From Table 5, it can be observed that our model surpassed other controlled versions and the baseline on all four domains. We notice that replacing our learnable novel-domain augmenter with hand-crafted random augmentations results in significant performance drops, and the performance was just marginally better than that of the baseline (i.e., *DeepAll*). This verifies that such random image transformations can merely achieve limited improvement on generalization capability. Although the average accuracy of *Jigsaw Puzzle* was better than that of *Random Augmentation* by about 2.7%, it was still worse than that of our full version by about 2.6%. This is possibly because that, while *Jigsaw Puzzle* provides more visual clues about spatial information as stated in [11, 12], there is no guarantee that such image transformation

Table 5: Ablation studies on PACS using ResNet-50 as the backbone.

| Module | Method | Photo | Art painting | Cartoon | Sketch | Average |
|---|---|---|---|---|---|---|
| Augmentation | DeepAll | 94.8 | 81.5 | 78.6 | 69.7 | 81.2 |
| | Random Aug. | 96.4 | 83.2 | 75.9 | 75.5 | 82.8 |
| | Jigsaw puzzle | 97.1 | 85.3 | 79.0 | **80.5** | 85.5 |
| Representation | Siamese archi. | 98.3 | 87.5 | 82.7 | 74.5 | 85.8 |
| | $F_S$ w/o EMA | 98.2 | 86.4 | 80.1 | 74.7 | 84.9 |
| | $F_S$ w/ EMA | 97.9 | 88.9 | 82.0 | 75.1 | 86.0 |
| | **Ours** $(G + F_T)$ | **98.9** | **90.0** | **83.5** | 80.0 | **88.1** |

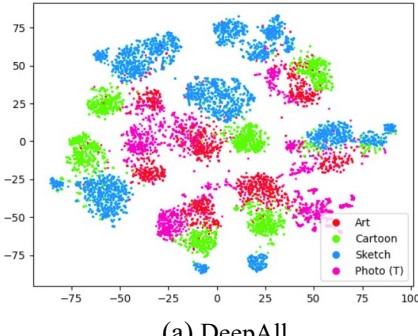

(a) DeepAll

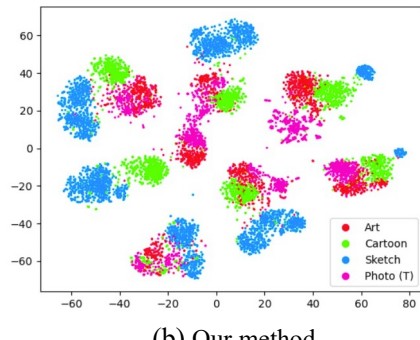

(b) Our method

Figure 2: t-SNE visualization on PACS with Photo as the unseen target domain. (a) Representations extracted by the baseline approach of DeepAll. (b) Representations derived by our approach.

would contribute to domain invariance. With the above experiments, our learnable novel-domain augmenter exhibits sufficient ability to generate novel-domain augmentations for facilitating the model robustness to unseen domains.

**Effectiveness of Domain Generalized Teacher**  From the results shown in the lower half of Table 5, we see that the performance dropped when we replace the Teacher-Student scheme by a *Siamese Architecture*, where the parameters are shared between the teacher and student networks. This is due to the fact that the Siamese architecture is prone to output collapsing solutions, hampering the derivation of domain generalized representations. In addition, we examine the performance of applying the trained student network to unseen domains instead of applying Teacher. *Student without EMA* denotes that Teacher is fixed during training, while *Student with EMA* denotes that Teacher is still updated with EMA which benefits the learning of Student. We observe that adopting EMA achieved the better results, but the performance of the above two versions (which apply Student) were still inferior to ours (which applies Teacher). From the above results, we confirm that Teacher updated with EMA would be less likely to be affected by possibly unrealistic domain augmentations during training, avoiding the degradation of its domain generalization capability. As verified by the above experiments, all components presented in our learning scheme would contribute to the domain generalization capability.

### 4.4.2 Visualization

We now qualitatively assess the ability of our approach in deriving domain generalizable features. As shown in Fig. 2, we apply t-SNE to compare the features $z$ derived by our teacher network $F_T$ with the features extracted by *DeepAll* network on PACS. In this figure, while the source image features extracted by *DeepAll* can be grouped according to their semantic categories, the target-domain features still cannot be properly separated. It can be observed that both source and target-domain features derived by our Teacher are sufficiently aligned, and the distances between different class clusters are more evident, indicating that equipped with our proposed adversarial teacher-student representation learning, our model is capable of learning more discriminative yet domain generalizable features.

Moreover, in Fig. 3, we visually compare the synthesized images by our method and those by the state-of-the-art data-generation method of DDAIG [7] using PACS as the training dataset. As described in Sec. 2, [7] learns to perturb the input images for confusing the domain classifier, with the goal of producing output images to be domain-agnostic. However, from Fig. 3, we see that images

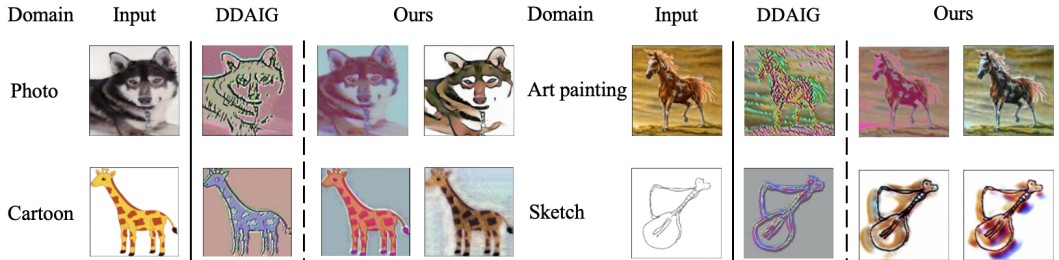

Figure 3: Visual comparisons of augmented novel-domain images produced by DDAIG [7] and ours on PACS dataset.

Table 6: Impact of momentum coefficient $\tau$ on Office-Home using ResNet-50 as the backbone.

| $\tau$ | Art | Clipart | Product | Real world | Average |
|---|---|---|---|---|---|
| 0.9 | 66.5 | 56.2 | 78.9 | 80.9 | 70.6 |
| 0.99 | 68.1 | 56.9 | 80.1 | 81.4 | 71.6 |
| 0.999 | **69.3** | **60.1** | **81.5** | **82.1** | **73.3** |

generated by DDAIG [7] tended to exhibit visual perturbation, which might not correspond to domain variations. On the contrary, our approach was capable of producing images in the data domains which are visually realistic yet distinct from source domains. We also note that, our model is trained in a deterministic manner, and the two augmented outputs are generated from our augmenter learned at different time steps with distinct mini-batch data sampled. This supports that our novel-domain augmentation mechanism is able to expand the training distributions.

### 4.4.3 Impact of the Momentum Coefficient $\tau$

In exponential moving average (EMA), $\tau$ is a momentum coefficient to control the update degree of our teacher network $F_T$. As shown in Table 6, we conducted ablation studies on Office-Home with ResNet-50 as backbone and observed that a large momentum coefficient $\tau$ by smoothly refining $\theta_T$ could achieve better performance than by rapid updating. These results indicate that a smooth refinement of Teacher avoids the degradation of generalization capability.

### 4.5 Generalization from A Single Source Domain

We evaluate our method on a more challenging DG task, single source domain generalization, to further verify the effectiveness of our method. In the single source DG setting, we only observe training data from a single source domain during training with the aim of generalizing to multiple unseen domains. To confirm that our approach can be extended to the single source DG setting, we conduct experiments on **PACS** and the large-scale benchmark dataset **DomainNet** with the ResNet-50 backbone. For PACS, we select *Photo* as the source domain and the remaining ones (i.e., *Art painting*, *Cartoon*, and *Sketch*) as the target domains. On the other hand, *Real* domain in DomainNet is chosen as the source domain, while *Clipart*, *Infograph*, *Painting*, *Quickdraw*, and *Sketch* domains serve as the target domains. We note that, since only a single source domain is observed during training, the margin $m$ in (4) is viewed as a hyperparameter instead of calculating from source domain data. Due to page limitation, additional experiments on PACS using *Art painting*, *Cartoon*, and *Sketch* as the single source domains are presented in the supplementary material.

We provide quantitative comparisons with the baseline (*DeepAll*), JiGen [11], and other three data-generation based methods [6, 7, 42] to evaluate the generalization capability on this challenging setting. As shown in Table 7, our approach performed favorably against the baseline (*DeepAll*) and the above DG methods on both benchmark datasets. It is worth noting that, compared with data-generation based methods of [6, 7, 42], our approach was able to achieve superior accuracy on all the target domains of interest. This confirms that, while our method can also be viewed as a data-generation based approach, we are able to better augment novel-domain data based on the observation of single source domain data. From the above experiments, the use of our approach for single source domain generalization tasks can be successfully verified.

Table 7: Single-source domain generalization on PACS and DomainNet using ResNet-50 as the backbone. Note that *Photo* of PACS and *Real* of DomainNet are selected as the single source domain for training.

| Method | PACS | | | | DomainNet | | | | | |
|---|---|---|---|---|---|---|---|---|---|---|
| | Art painting | Cartoon | Sketch | Average | Clipart | Infograph | Painting | Quickdraw | Sketch | Average |
| DeepAll | 60.7 | 23.5 | 29.0 | 37.7 | 34.5 | 15.7 | 40.7 | 3.6 | 25.9 | 24.1 |
| JiGen [11] | 63.6 | 28.5 | 30.2 | 40.8 | 50.0 | 19.0 | 46.3 | 7.2 | 35.5 | 31.6 |
| CrossGrad [6] | 64.2 | 29.4 | 32.1 | 41.9 | 49.4 | 19.3 | 47.3 | 5.8 | 35.6 | 31.5 |
| DDAIG [7] | 64.1 | 32.5 | 29.6 | 42.1 | 41.4 | 16.5 | 40.9 | 3.2 | 26.7 | 25.7 |
| M-ADA [42] | 64.6 | 34.6 | 26.6 | 41.9 | 50.3 | 19.5 | 48.1 | 7.1 | 36.0 | 32.2 |
| **Ours** | **68.2 ± 0.9** | **36.3 ± 0.9** | **33.5 ± 0.3** | **46.0** | **52.2 ± 0.3** | **21.6 ± 0.2** | **50.1 ± 0.2** | **8.1 ± 0.3** | **38.3 ± 0.4** | **34.1** |

## 5 Conclusion

In this paper, we proposed Adversarial Teacher-Student Representation Learning for addressing domain generalization classification tasks. By alternating between the training stages of Teacher-Student representation learning and novel-domain augmentation in an adversarial manner, our learning scheme allows derivation of domain generalizable representations while semantic information properly preserved. We conduct extensive experiments, including comparisons to state-of-the-art meta-learning and data-generation based DG methods and ablation studies, which quantitatively and qualitatively confirm the effectiveness and robustness of our proposed approach in solving DG classification by training on single or multiple source-domain data.

**Acknowledgement**  This work is supported in part by the Ministry of Science and Technology of Taiwan under grants MOST 110-2634-F-002-036 and 110-2221-E-002-121.

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
