# Adversarial Teacher-Student Representation Learning for Domain Generalization Supplementary Material

**Fu-En Yang**[1,2]  **Yuan-Chia Cheng**[1]  **Zu-Yun Shiau**[1]  **Yu-Chiang Frank Wang**[1,2]

[1]Graduate Institute of Communication Engineering, National Taiwan University, Taiwan
[2]ASUS Intelligent Cloud Services, Taiwan
{f07942077, r08942154, r09942069, ycwang}@ntu.edu.tw

## A    Pseudo-Code of Adversarial Teacher-Student Representation Learning

We provide the pseudo-code of our Adversarial Teacher-Student Representation Learning in Algorithm 1.

---

**Algorithm 1:** Adversarial Teacher-Student Representation Learning

**Input:** Number of iterations $N_{iter}$, number of warm up iterations $N_{warm}$, learning rate $\gamma$,
  Teacher $F_T$, Student $F_S$, novel-domain augmenter $G$ and classifier $C$
**Data:** $N$ source domains $\mathcal{D}_{tr} = \{\mathcal{D}_1, \mathcal{D}_2, ..., \mathcal{D}_N\}$
**Output:** Teacher $F_T$

1  **for** $i$ in $1 : N_{iter}$ **do**
2     Randomly sample a minibatch $(x, y)$ from source domains ;
3     **if** $i < N_{warm}$ **then**
4        Update $F_T$ and $C$ with $\mathcal{L}_{ce}(C(F_T(x)), y)$;
5     **else**
6        **Domain Generalized Representation Learning**
7        $\tilde{x} = G(x)$;
8        $z = F_T(x)$, $\tilde{z} = F_S(\tilde{x})$;
9        Compute $\mathcal{L}_{dis}^F$ (Eq.1) and $\mathcal{L}_{ce}(C(\tilde{z}), y)$;
10       Update $F_S$ via back propagation. $\theta_S \leftarrow \theta_S - \gamma \frac{\partial(\mathcal{L}_{dis}^F(z,\tilde{z}) + \mathcal{L}_{ce}(C(\tilde{z}),y))}{\partial \theta_S}$ (Eq.2);
11       Update $F_T$ via EMA. $\theta_T \leftarrow \tau\theta_T + (1 - \tau)\theta_S,$  where $\tau \in [0, 1)$ (Eq.3);
12       **Novel Domain Augmentation**
13       Compute $\mathcal{L}_{dis}^G$ (Eq.4) and $\mathcal{L}_{ce}(C(\tilde{z}), y)$;
14       Update $G$ via back propagation. $\theta_g \leftarrow \theta_g - \gamma \frac{\partial(-\mathcal{L}_{dis}^G(z,\tilde{z}) + \mathcal{L}_{ce}(C(\tilde{z}),y))}{\partial \theta_g}$ (Eq.5);
15    **end**
16 **end**

---

## B    Further Quantitative Comparisons

For multiple source domain generalization, in addition to PACS, Office-Home, and DomainNet datasets included in main paper, we further evaluate the effectiveness of our approach on two benchmark datasets, Digits-DG and VLCS.

35th Conference on Neural Information Processing Systems (NeurIPS 2021).

Table 1: Comparisons to existing methods on Digits-DG in leave-one-domain-out settings. **Bold** denotes the best result.

| Target | DeepAll (baseline) | CCSA [8] | MMD-AAE [9] | CrossGrad [10] | DDAIG [7] | L2A-OT [11] | MixStyle [12] | Ours |
|---|---|---|---|---|---|---|---|---|
| MNIST | 95.3 | 95.2 | 96.5 | 96.7 | 96.6 | 96.7 | 96.5 | **97.9** $\pm$ 0.1 |
| MNIST-M | 61.1 | 58.2 | 58.4 | 61.1 | **64.1** | 63.9 | 63.5 | 62.7 $\pm$ 0.3 |
| SVHN | 62.3 | 65.5 | 65.0 | 65.3 | 68.6 | 68.6 | 64.7 | **69.3** $\pm$ 0.1 |
| SYN | 79.5 | 79.1 | 78.4 | 80.2 | 81.0 | 83.2 | 81.2 | **83.7** $\pm$ 0.2 |
| Average | 74.5 | 74.5 | 74.6 | 75.8 | 77.6 | 78.1 | 76.5 | **78.4** |

Table 2: Comparisons to existing methods on VLCS using AlexNet in leave-one-domain-out settings. **Bold** denotes the best result.

| Target | DeepAll (baseline) | MMD-AAE [9] | MLDG [13] | Epi-FCR [14] | JiGen [15] | MASF [16] | MetaVIB [17] | EISNet [18] | RSC [19] | Ours |
|---|---|---|---|---|---|---|---|---|---|---|
| PASCAL | 66.3 | 67.7 | 67.7 | 67.1 | 70.6 | 69.1 | 70.3 | 69.8 | 73.9 | **76.9** $\pm$ 0.4 |
| LabelMe | 61.4 | 62.6 | 61.3 | 64.3 | 60.9 | **64.9** | 62.7 | 63.5 | 61.9 | 62.9 $\pm$ 0.7 |
| Caltech | 97.2 | 94.4 | 94.4 | 94.1 | 96.9 | 94.8 | 97.4 | 97.3 | **97.6** | 97.2 $\pm$ 0.1 |
| Sun | 68.1 | 64.4 | 64.4 | 65.9 | 64.3 | 67.6 | 67.9 | 68.0 | 68.3 | **69.6** $\pm$ 0.3 |
| Average | 73.3 | 72.3 | 72.3 | 72.9 | 73.2 | 74.1 | 74.5 | 74.7 | 75.4 | **76.1** |

## B.1 Datasets

**Digits-DG** consists of four domains, MNIST [1], MNIST-M [2], SVHN [3] and SYN [2], with digit images of varying font styles and background colors. Each domain contains 10 categories, with 6000 images in total. Images are divided into the training split and the validation split at a ratio of 8:2. **VLCS** [4] is a domain generalized visual classification benchmark, which includes five categories from four domains (PASCAL VOC 2007, LabelMe, Caltech, and Sun datasets), with the domain gap mainly from camera viewpoints, types of camera, or illumination conditions, etc. Images are divided into the training split and the validation split at a ratio of 9:1.

## B.2 Implementation Details and Results

For Digits-DG, input images are resized to $32 \times 32$ pixels, and the backbone of $F_T$ and $F_S$ consists of four convolution layers, with the kernel size 3 and channel size 64. Each convolution layer is followed by a ReLU and a maxpooling layer with the kernel size 2. Classifier $C$ is realized by a fully-connected layer, and maps a flattened feature vector to a 10 dimensional output. $F_S$ is trained with SGD, initial learning rate of 0.05 and batch size of 128 for 60 epochs. For VLCS, input images are resized to $224 \times 224$ pixels, and we use AlexNet [5] pre-trained on ImageNet [6] as the backbone of our teacher and student networks. $F_S$ is trained with the SGD optimizer, with an initial learning rate of 0.0005, and a batch size of 32 for 60 epochs. The learning rate is decayed by 0.1 after 30 epochs. For both Digits-DG and VLCS, $F_T$ is updated via EMA with the momentum coefficient $\tau$ of 0.999 by default. Our novel-domain augmenter $G$ is realized by a fully convolutional network similar to the generator's architecture in [7] and trained with the SGD optimizer.

Tables 1 and 2 show the quantitative comparisons with existing DG methods on Digit-DG and VLCS, respectively. Our approach still achieved satisfactory performance over the state-of-the-art models on all domains, with the reported highest average accuracy on both Digits-DG (**78.4**%) and VLCS (**76.1**%). The above experimental results further support the effectiveness and robustness of our method to tackle domain generalized visual classification tasks.

## B.3 Generalization from A Single Source Domain

We conduct additional experiments with the ResNet-50 backbone on PACS using *Art painting*, *Cartoon*, and *Sketch*, respectively, as the single source domain to further confirm the use of our method to deal with such a challenging setting. As shown in Tables 3, 4, and 5, our approach performed favorably against the baseline (*DeepAll*) and the existing DG methods regardless of the source domain we selected. The above quantitative experiments thus confirmed the effectiveness and the domain generalization ability of our proposed model.

Table 3: Single source domain generalization on PACS using ResNet-50 as the backbone. Note that *Art painting* of PACS is selected as the single source domain for training.

| PACS-A | Photo | Cartoon | Sketch | Average |
|---|---|---|---|---|
| DeepAll | 96.9 | 57.0 | 42.8 | 65.6 |
| JiGen [15] | 96.3 | 61.4 | 52.7 | 70.1 |
| CrossGrad [10] | 97.3 | 62.5 | 45.9 | 68.6 |
| DDAIG [7] | 97.0 | 61.5 | 54.1 | 70.9 |
| M-ADA [23] | 97.2 | 63.7 | 47.0 | 69.3 |
| **Ours** | **97.4** $\pm$ 0.1 | **64.0** $\pm$ 0.4 | **56.1** $\pm$ 0.2 | **72.5** |

Table 4: Single source domain generalization on PACS using ResNet-50 as the backbone. Note that *Cartoon* of PACS is selected as the single source domain for training.

| PACS-C | Photo | Art painting | Sketch | Average |
|---|---|---|---|---|
| DeepAll | 87.0 | 64.0 | 55.8 | 68.9 |
| JiGen [15] | 87.1 | 65.3 | 66.3 | 72.9 |
| CrossGrad [10] | 86.8 | 66.4 | 65.4 | 72.9 |
| DDAIG [7] | 86.8 | 68.5 | 65.9 | 73.7 |
| M-ADA [23] | **87.7** | 67.7 | 63.1 | 72.8 |
| **Ours** | 87.6 $\pm$ 0.2 | **70.0** $\pm$ 0.1 | **68.4** $\pm$ 0.8 | **75.3** |

Table 5: Single source domain generalization on PACS using ResNet-50 as the backbone. Note that *Sketch* of PACS is selected as the single source domain for training.

| PACS-S | Photo | Art painting | Cartoon | Average |
|---|---|---|---|---|
| DeepAll | 25.1 | 21.0 | 43.7 | 29.9 |
| JiGen [15] | 37.5 | **37.1** | 55.6 | 43.4 |
| CrossGrad [10] | 31.6 | 25.5 | 49.1 | 35.4 |
| DDAIG [7] | 28.6 | 30.0 | 59.3 | 39.3 |
| M-ADA [23] | 26.0 | 23.1 | 52.0 | 33.7 |
| **Ours** | **39.2** $\pm$ 0.2 | 31.0 $\pm$ 0.3 | **61.0** $\pm$ 0.2 | **43.7** |

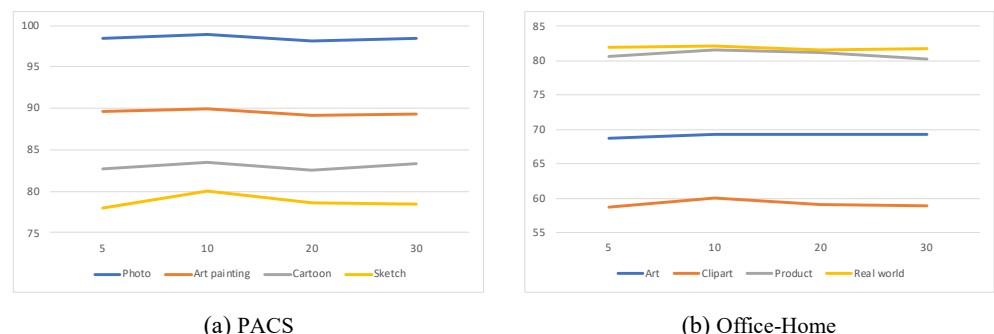

(a) PACS         (b) Office-Home

Figure 1: Impact of the epoch of warm-up stage on (a) PACS and (b) Office-Home using ResNet-50 as the backbone. Note that x and y axes denote the epoch of warm-up stage and top-1 classification accuracy (%), respectively.

Moreover, we provide additional comparisons of single-source domain generalization on Digit datasets. Following two very recent SOTAs of [20, 21] using MNIST [1] as the single source domain, our method showed promising results 61.7%, 83.2%, 69.3%, and 87.4% on SVHN [3], MNIST-M [2], SYN [2], and USPS [22], respectively. We achieved an average accuracy of 75.4% and outperformed [20] and [21] which resulted in 74.8% and 61.3%, respectively. And, as discussed in Section 2, PDEN [20] has a much larger memory requirement and a more complex model (i.e., more hyperparameters to select). The above additional experiments further confirm the effectiveness of our method for single-source domain generalization.

### B.4 Parameter Analysis

**Impact of the Warm-up Stage** In this sub-section, we conduct a detailed analysis of the impact of our warm-up stage on both PACS and Office-Home with ResNet-50 as the backbone. As shown in Fig. 1, the performance does not exhibit drastic fluctuations despite using different warm-up epochs, further showing that our model is stable and robust and that the training epoch of warm-up stage is not the most influential factor to the result. We eventually choose 10 epochs for warm-up training as default in our experiments.