# OpenReview forum: "Adversarial Teacher-Student Representation Learning for Domain Generalization"
_NeurIPS.cc/2021/Conference — NeurIPS 2021 Spotlight_

### Official Review · Reviewer_oJ4V · 2021-06-27

**Rating:** 7
**Confidence:** 5

**Summary:**

This paper proposes a new data augmentation method for domain generalization. The main idea is to train a neural network-based generator to generate novel images that can be correctly classified by a student model and in the meantime, make the feature discrepancy larger between the student model and the EMA teacher model. The student model, which is fed with the generated images, is trained to minimize both the classification error and the feature discrepancy with its teacher. Only the EMA teacher is deployed at test time. This method is evaluated on both multi- and single-source domain generalization settings, outperforming previous SOTAs by a clear margin.

**Limitations And Societal Impact:**

The authors have properly discussed the limitations of this work in Sec.2, but do not talk about potential negative societal impact. Since this work is a data augmentation method based on learning a generator network from data, the learning process might be biased toward specific datasets used.

**Main Review:**

**Strength**

- Though learning-based data augmentation has been studied in domain generalization, the idea of using the teacher-student paradigm proposed in this work is novel and could add value to this line of research.

- The evaluation is comprehensive and sufficient, which covers both the multi- and single-source settings as well as different network architectures. The results are encouraging (clearly better than previous data augmentation methods).

**Weakness**

- Section 2 lacks discussions on the differences between this work and existing data augmentation methods. In addition, I do not see a strong connection between this work and SSL methods—except those based on multi-view learning (contrastive learning).

- It's unclear why the single-layer classifier is trained and then fixed during the entire learning process. Why not train the classifier along with the student model?

- The loss formulation in Eq.4 is a bit strange. Once the feature discrepancy is smaller than the margin, no gradients will be backpropagated due to the max(x, 0) function. Why not simply maximize Eq.1 for maximizing the feature discrepancy?

- Compared with confusing a domain classifier or maximizing a distribution distance for synthesizing novel-domain images, the design of maximizing the feature discrepancy between a model and its EMA is not that intuitive. More insights should be provided to explain how this design helps the generator produce images that deviate from the source distributions.

- Line217 says that the generator's architecture is similar to that used in CycleGAN. I'd like to confirm if this generator is trained with only the classification loss and the feature discrepancy loss? Did the authors use other losses like the L1 loss to train the generator?

- In Fig.3, why there are two generated images given the same input? Isn't the generator a deterministic model?

- Line218: Why pretrain the EMA teacher model while giving a random initialization to the student model?

- Line290: What is the difference between Student with EMA and the proposed model (G + F_T)?

---
Post-rebuttal update:

I have gone through the authors' responses, which have addressed my concerns. I have read other reviewers' comments as well.

The proposed data augmentation approach is novel in DG and well justifed with extensive experiments and ablation studies. My concerns raised during the initial review were mainly related to the presentation and clarity, which have been cleared in the rebuttal. I think this paper is worth sharing to the DG community so I change my rating to clear accept.

**Time Spent Reviewing:**

3.5

---

> ### Author Response · Authors · 2021-08-10
> **Response to Reviewer oJ4V**
>
> **1. Section 2 lacks discussions on the differences between this work and existing data augmentation methods. In addition, I do not see a strong connection between this work and SSL methods—except those based on multi-view learning (contrastive learning).**
>
> We thank the reviewer for pointing this out, and we are more than happy to clarify this issue. The existing data augmentation methods focus on either generating perturbed images to confuse the domain classifier [6, 7] or to maximize the distribution distance [8], or mixing the feature statistics across source domains [9]. However, such perturbation or the interpolated features from source domains would not necessarily exhibit sufficient domain generalization ability. As for our method, we perform data augmentation in the image level, under the guidance of feature discrepancy observed across data domains and with classification objectives jointly observed. We compare our method with works mentioned above in Table 3, 4 and Fig. 3.
>
> The connection between our method and SSL is implied by the teacher-student like SSL networks (e.g., MoCo [16] and BYOL [17]), which share the same goal of deriving robust yet semantically similar representations from data and its variants. We also note that some works have extended SSL into the supervised learning regime (e.g., [A] which applies the contrastive objective for supervised pretraining), which is similar to our incorporating SSL techniques into DG (also a supervised learning).
>
> We will add the above discussions into Section 2, which makes our discussions more complete.
>
>
>
> **2. It's unclear why the single-layer classifier is trained and then fixed during the entire learning process. Why not train the classifier along with the student model?**
>
> We thank the reviewer for raising this issue. We only pretrain the classifier using source domain data available, and keep it fixed during the learning of our teacher-student augmentation framework. If we allow the update of this classifier during the training process, it might observe undesirable outputs and affect the learning of both augmenter and teacher/student modules in the early training stage, where either the augmented data or its extracted features is not yet quality. Although we have already made remarks in L164-165 and L291-293, we will add the above additional discussions into our future/final version to make this clearer.
>
> **3. The loss formulation in Eq.4 is a bit strange. Once the feature discrepancy is smaller than the margin, no gradients will be backpropagated due to the max(x, 0) function. Why not simply maximize Eq.1 for maximizing the feature discrepancy?**
>
> We thank the reviewer for pointing this up. It is indeed an error, and we provide the corrected version below. As you can see, the subscript of the plus sign should be a minus sign, which only outputs non-zero values if the feature discrepancy is smaller than the margin.
> $$
> \max_{G}\mathcal{L}_{dis}^G(z,\tilde{z})=[\left \| {\left \| \frac{z}{\left \|| z \right \||_2}-\frac{\tilde{z}}{\left \|| \tilde{z} \right \|_2} \right \|}  \right \|^2_2-m]_\-=[\left \| {\left \| \frac{F_T(x)}{\left \|| F_T(x)) \right \||_2}-\frac{F_S(G(x))}{\left \|| F_S(G(x)) \right \||_2} \right \|} \right \|^2_2-m]_\-,
> $$
> where $[\cdot]_\-=\min(\cdot,0)$.
>
> While our implementation is correct, we are sorry for the confusion and sincerely thank the reviewer for the correction.
>
> **4. Compared with confusing a domain classifier or maximizing a distribution distance for synthesizing novel-domain images, the design of maximizing the feature discrepancy between a model and its EMA is not that intuitive. More insights should be provided to explain how this design helps the generator produce images that deviate from the source distributions.**
>
> We thank the reviewer for raising this issue. We note that, confusing a domain classifier has been widely applied to domain adaptation works yet only allows derivation of features invariant to existing source domains. Thus, when applying such learning strategies to domain generalization, the recognition/generation ability to unseen domain data might be limited (as confirmed in Table 1, 3 when comparing to [1, 7]).
>
> As pointed out by Reviewer 5PA5, both a recent CVPR 2021 work [B] and we avoid the above limitation and share the same idea of learning to augment data for unseen domains. More specifically, we deploy a novel-domain augmenter, which is learned to augment data with sufficient cross-domain feature discrepancy, followed by teacher-student modules for learning domain-generalizable features. With class labels jointly observed during training, we are able to augment images in out-of-source distributions yet with desirable class information.
>
> We note that, while our student module is directly updated by the augmented data, applying EMA in our learning framework would alleviate the problem of resulting in the teacher module from directly observing unrealistic domain augmentations (see degraded results by our model without EMA in Table 5). [20, 16, 21] also utilize the same strategy to stabilize the training of their teacher models. We will be happy to add the above discussions into Section 2 for making this issue clearer.
>
>
>
> **5. Is this generator trained with only classification and feature discrepancy losses? Did you use other losses like L1 loss to train the generator?**
>
> Yes, we train our generator with only classification and feature discrepancy losses, which not only update our teacher-student modules but also the augmenter. In other words, we did not consider L1 or other losses to train our augmenter.
>
> **6. In Fig.3, why are two generated images given the same input? Isn't the generator a deterministic model?**
>
> We apologize for the possible confusion. The visualization examples shown in Fig. 3 (i.e., the two augmented outputs are generated from our augmenter learned at different time steps with distinct mini-batch data sampled). Our model is indeed a deterministic one.
>
> **7. Line218: Why pretrain the EMA teacher model while giving a random initialization to the student model?**
>
> In our teacher-student training scheme, we pretrain the teacher module using on source-domain data for model initialization, which serves as a reliable guidance to train the student module. On the other hand, the aim of the student module is to observe novel-domain augmentation and then distill the knowledge to gradually refine the teacher module. Thus, for training the student module, either random initialization or initialization from the teacher (like some SSL works) would be feasible. To further verify this claim, we performed additional experiments on PACS using the above two strategies, and we observed negligible differences (85.3% with random initialization, and 84.9% with model initialized by teacher).
>
> **8. Line290: What is the difference between Student with EMA and the proposed model (G + F_T)?**
>
> We are sorry for the confusion. Student with EMA denotes that the student network is utilized during inference, while G + F_T represents the full version of our model, which applies the learned teacher network for inference. We will revise the notation in Table 5 and avoid possible confusion.
>
> \
> [A] - Supervised Contrastive Learning. NeurIPS 2020
>
> [B] - Progressive Domain Expansion Network for Single Domain Generalization. CVPR 2021

---

> > ### Comment · Reviewer_oJ4V · 2021-08-11
> > **Solid response; concerns resolved**
> >
> > > We only pretrain the classifier using source domain data available, and keep it fixed during the learning of our teacher-student augmentation framework. If we allow the update of this classifier during the training process, it might observe undesirable outputs and affect the learning of both augmenter and teacher/student modules in the early training stage, where either the augmented data or its extracted features is not yet quality.
> >
> > Please add the relevant discussion to the paper because such a design is unusual and needs further clarification.
> >
> > > We apologize for the possible confusion. The visualization examples shown in Fig. 3 (i.e., the two augmented outputs are generated from our augmenter learned at different time steps with distinct mini-batch data sampled). Our model is indeed a deterministic one.
> >
> > Perhaps use a couple of sentences to clarify this in the caption or text.

---

> > > ### Author Response · Authors · 2021-08-12
> > > **Follow-up comment**
> > >
> > > Definitely. We thank the reviewer for the reminder, and we will add the above responses on implementation details and visualization examples (in deterministic not generative ways) into our revised version for improved clarification.

---

### Official Review · Reviewer_3Rjy · 2021-07-16

**Rating:** 7
**Confidence:** 4

**Summary:**

The article proposes an approach for domain generalization (DG) based on adversarial augmentation and knowledge distillation. In particular, the model contains three components: a teacher, a student, and an augmenter. The student receives input processed by the augmenter while the teacher the original data, and a discrepancy loss measures the difference between student and teacher features for a given sample. Training is held out in two cyclic stages: in the first, the augmenter is fixed and the student parameters are learned by minimizing the semantic and discrepancy loss, with the teacher being an exponential moving average of the student. In the second stage, student and teacher are fixed and the augmenter is updated by minimizing the semantic classification loss and maximizing the discrepancy one. The teacher is used at test time. Experiments on a variety of benchmarks show the efficacy of the proposed approach.

**Limitations And Societal Impact:**

No limitations are discussed as well as possible negative social impacts. While I deem the latter to not be present, it would have been important to discuss the limits of the work in terms of, e.g. variety of augmentations produced (Figure 3 shows mostly color-based transformations) and future improvements of the single components.

**Main Review:**

Overall:
The paper proposes a relatively simple yet effective approach for domain generalization, merging the strength of Mean-Teacher [20] with adversarial data augmentation [8], showing non-marginal improvements over the state of the art in multiple datasets and in both single and multi-source domain generalization. Since both distillation and augmentation strategies are new in the context of domain generalization and I did not spot any major weakness I lean toward the acceptance of the manuscript. Below I detail my comments.

Strengths:
+ Exploiting the consistency between the teacher and the student predictions to train the data augmentation network is interesting since 1) standard DG approaches for domain generalization use a domain classifier (e.g. [7,8]); 2) does not require any domain information; 3) exploits the same network used to predict the semantic of the input, being computationally less demanding.

+ Not many multi-source DG models show to be effective in the single-source DG task (one example is [28]). The proposed approach outperforms existing generative and non-generative approaches in both settings, sometimes by a margin (e.g. +~2% on the challenging single-source DG in DomainNet, Table 2 and +1.2% on multi-source DG in Office-Home, Table 4).

+ The ablation study (Table 5) thoroughly analyzes the impact of each component and design choice.

Weaknesses:

1. The article does not fully specify some of the methodological components, even if it is easy to understand their role. In particular, the classification loss is indicated in Figure 1 but briefly mentioned in the text (line 183).

2. The organization of the experiments can be improved. For instance, multiple tables are used for the comparison with the state of the art, containing redundant results, with the main difference being on the competitors (e.g. Table 1-3 and Table 2-4). It would be good to keep one table for each tested dataset while specifying which methods are generative and which are not. The saved space can be used to either integrate other analyses from the supplementary (e.g. B3, C) or add more qualitative augmentation results.

**Time Spent Reviewing:**

4

---

> ### Author Response · Authors · 2021-08-10
> **Response to Reviewer 3Rjy**
>
> **1. The article does not fully specify some of the methodological components, even if it is easy to understand their role. In particular, the classification loss is indicated in Figure 1 but briefly mentioned in the text (line 183).**
>
> We thank the reviewer for the suggestion, and we are happy to clarify our model design and loss functions. As noted in L183, our classification objective considers a standard cross-entropy loss. As for the other (and remaining) objective, we consider the feature discrepancy loss for deriving domain-generalized features and enforcing out-of-source distribution augmentation, which are defined in Equations 1 and 4, respectively.
>
> **2. The organization of the experiments can be improved. It would be good to keep one table for each tested dataset while specifying which methods are generative and which are not. The saved space can be used to either integrate other analyses from the supplementary or add more qualitative augmentation results.**
>
> We thank the reviewer for this useful suggestion. We will make proper adjustments accordingly in our final version. We also note that only methods of [6, 7, 8, 9] are generative models.
>
> Once again, we thank Reviewer 3Rjy for providing the suggestions and pointing out possible limitations of our work. We will make adjustments as suggested in our revised version.

---

> > ### Comment · Reviewer_3Rjy · 2021-08-20
> > **Thanks for the response**
> >
> > I thank the authors for the extensive feedback on my and the other reviewers' concerns. After reading the response and the other reviews I am still convinced of the value of the work, thus I confirm my initial positive rating.
> >
> > I would still suggest the authors expand the qualitative results (see point 2) in the main manuscript.

---

> > > ### Author Response · Authors · 2021-08-22
> > > **Follow-up**
> > >
> > > We thank the reviewer again for the positive feedback. We will definitely make our qualitative results more complete as suggested (e.g., reorganize the quantitative results, and thus move more qualitative ones from the supplementary to the main manuscript). We sincerely appreciate your valuable suggestions.

---

### Official Review · Reviewer_kqaD · 2021-07-16

**Rating:** 6
**Confidence:** 4

**Summary:**

This paper proposes Adversarial Teacher-Student Representation Learning to learn domain invariant features for domain generalization. The method learns teacher-student co-training and novel-domain data augmentation in an adversarial manner.

**Limitations And Societal Impact:**

Yes, they mention it in section 2.

**Main Review:**

The paper is well-written and easy to understand.

Authors proposes the data augmentation to generate synthetic data for domain generalization problem. Such data augmentation is used in domain adaptation and few shot classification often and it is interesting that authors incorporate this idea with domain generations.

The formulation of minimizing the data discrepancy when training the teacher-student network and maximizing the data discrepancy when updating the data augmenter. Authors evaluate their methods on multiple domain adaptation dataset collections with different network backbones. For the data augmentation network G, authors proposes to augment the input image which requires a large augmentation network. But augmenting extracted features instead of the input image is widely used in data hallucination [A], [B]. It is unclear of the reason why authors choose to augment the image.

The main experiments shows the proposed methods achieve better average acc in domain generalization. Ablation studies show the effectiveness of adversarial augmenter and domain generalized teacher. Visualization provides some generated images. It is shown that the data augmenter generates re-styled images based on the input. The re-styling is visually similar to the re-coloring instead of content preserved style transfer [C]. I am not sure with this kind of data augmentation whether the augmenter succeeds in generating out-of-distribution data and benefits the generalization.

Some questions:
1. Why do you choose to augment the image rather than features?
2. Instead of re-coloring image, is there any other augmentation generated by the augmentor?
3. Is there any noise as input the augmentor in order to generate multiple outputs with the same image?

[A] Wang, Yu-Xiong, et al. "Low-shot learning from imaginary data." Proceedings of the IEEE conference on computer vision and pattern recognition. 2018.

[B] Pahde, Frederik, et al. "Discriminative hallucination for multi-modal few-shot learning." 2018 25th IEEE International Conference on Image Processing (ICIP). IEEE, 2018.

[C] Huang, Xun, et al. "Multimodal unsupervised image-to-image translation." Proceedings of the European conference on computer vision (ECCV). 2018.

Update: Thanks for the authors' response. It addresses my concerns described above and explains very clearly the difference between FSL hallucinations and DG's. I think it is good to include such discussion in the final version, either the main paper or supplementary. I updated the rating.

**Time Spent Reviewing:**

4hr

---

> ### Author Response · Authors · 2021-08-10
> **Response to Reviewer kqaD**
>
> **1. Why do you choose to augment images rather than features?**
>
> We thank the reviewers for giving us the opportunity to clarify this issue. Since DG aims at tackling unknown domain shifts, a proper data augmentation strategy is to produce data with out-of-source distributions via diversifying training data domains. And, since source-domain data typically share the same label set, the above domain diversities would be style variance which can be generally observed in the image level (as [7, 8] and ours do).
>
> On the other hand, augmenting or hallucinating feature-level data is preferable for FSL works like [A, B]. Different from DG tasks, such FSL models require to hallucinate data with sufficient intra-class variations (not data domain variations). Thus, manipulating data in the feature level would be typically considered for FSL tasks.
>
> [A] Wang, Yu-Xiong, et al. "Low-shot learning from imaginary data." CVPR 2018.
>
> [B] Pahde, Frederik, et al. "Discriminative hallucination for multi-modal few-shot learning." ICIP 2018.
>
> **2. Instead of re-coloring images, are there any other augmentations generated by the augmentor?**
>
> We thank the reviewer for giving us the opportunity to justify this issue. As shown in the first example in Fig. 3, the second image augmented from the Photo domain exhibits watercolor style with blurred and simplified boundaries. As for the second output augmented from Sketch, it is an art-painting like image with softer strokes. Thus, our augmenter introduces image domain varieties and performs more than image recoloring. This also supports why the use of our trained model would better tackle the DG classification tasks (e.g., Table 3 in which we achieved 85.3% while SOTA augmentation-based DG approaches of [7, 8] reported 83.1% and 82.8%).
>
> **3. Is there any noise as input to the augmentor in order to generate multiple outputs with the same image?**
>
> No, as depicted in Fig. 1 and discussed in L216-217, our augmenter is a deterministic model and is implemented by an FCN (as suggested by [24]). Thus, we do not take noise as additional inputs. We apologize for possible confusion with the visualization examples shown in Fig. 3, in which the two augmented outputs are generated from our augmenter learned at different time steps with distinct mini-batch data sampled.

---

### Official Review · Reviewer_5PA5 · 2021-07-19

**Rating:** 7
**Confidence:** 4

**Summary:**

The paper studies the problem of Domain Generalization (DG), where a model trained on single or multiple source distributions / domains is expected to generalize to data from unseen / novel distributions at test-time. The authors propose an adversarial teacher-student representation learning setup consisting of two stages -- (1) synthesizing “plausible” out-of-domain instances as augmented source data (termed Novel Domain Augmentation) and (2) enforcing the representations obtained from augmented and vanilla source domain instances to be similar and relevant to the task at hand (termed Domain Generalized Representation Learning). This is achieved via teacher (fed regular instances) and student (fed augmented instance) encoders and a novel data augmentor unit. In the second stage, the student encoder is incentivized to be good at the downstream task and ensure representations from augmented samples are similar to the vanilla ones (the teacher parameters are kept up to date via an EMA step). To generate novel augmented samples, the augmentor unit is incentivized to maximize the discrepancy between the representations obtained from the teacher and student encoders. Broadly, the proposed approach falls in the category of DG relying on hallucinating reasonable out-of-domain samples (without assuming any access to domain labels) during training. Obtained results indicate that the proposed approach is competitive with existing DG approaches.

**Ethical Concerns:**

Not applicable.

**Ethics Review Area:**

["I don’t know"]

**Limitations And Societal Impact:**

The strengths and weaknesses highlighted in the review form the basis of my rating. I think addressing the points highlighted under weaknesses, especially the clarification and experimental concerns will help improve the paper. I would encourage the authors to do the same.

I don't see any potential negative social impacts of the work.

**Main Review:**

I will now highlight the strengths and weaknesses of the submission, which will address quality, clarity, originality and significance of the submission.

With the exception of the points covered under weaknesses, I believe the following are the strengths of the submission in its current form:

- The paper is generally well-written and easy to follow. With the exception of a few points under weaknesses regarding clarity issues, the authors do a decent job of motivating several components of the proposed approach -- the multi-view perspective on Domain Generalization, the Adversarial Teacher Student setup and other design choices.

- From a modelling standpoint, perhaps one of the biggest strengths of the paper lies in the fact that it does not rely on domain-labels -- making it useful for not only the multi-source but also the single-source domain generalization (DG) setup. More importantly, there is value on relying on DG approaches agnostic to domain labels especially for settings where access to domain labels may be a restrictive assumption (due to annotation cost / lack of clarity in terms of what constitutes different domains -- for instance, datasets like CARLA, CityScapes, etc.).

- The proposed approach seems simple, intuitive and results in competitive (on PACS) and improved (on Office-Home) multi-source DG performance. In terms of single-source DG performance, the proposed approach seems to offer improvements on both PACS and DomainNet (under the experimental settings considered; see weaknesses for more details.). In addition to competitive quantitative results, the ablations in section 4.4.1 demonstrate the utility of different components in the proposed approach.

- The simplicity, intuitiveness and consistent improvements offered by the proposed approach are likely going to make the submission useful for future work to build on top of (provided some of the experimental comparisons are exhaustive; see weaknesses). Overall, I believe this, combined with the lack of reliance on domain labels generally makes the approach valuable.

I will now highlight major weaknesses associated with the draft below. These are centered mostly around some clarity issues and lack of some experimental comparisons.

 - L178-179 states that the margin m (an important component in the proposed approach) in the objective used to update the novel domain augmentor is “calculated by the L2 distance between the means of the sampled source domains”. It’s unclear to me what this means -- is this the average L2 distance b/w the means across all possible pairs of source domains in the mini-batch? If not, could the authors comment on how exactly this is computed for the multi-source case? Similarly, in the single source generalization setting, m is treated explicitly as a hyper-parameter. However, it’s unclear how the best m is chosen for the single source experiments. Is the best m chosen based on target domain performance? If not, could the authors comment on how this is chosen?

- Right now, there’s only one line in the paper (and supplementary) detailing how the novel-domain augmentor is realized (as an image translation FCN network) and is hard to locate. The paper would benefit from discussing the architecture and pre-training details (if any) of G earlier section 3. Did the authors experiment with other generators as choices for the augmentor design? Including a discussion about the choice of augmentor architecture (image translation networks vs other plausible generators) would further benefit the paper.

- While (a) Table 5 clearly highlights the progressive improvements offered by the novel domain augmentor + the teacher-student (EMA) representation learning scheme over RandAugment and JigSaw and (b) Tables 3 & 4 highlight improvements over other data-generation DG approaches, the paper would benefit from discussing recent work [A] (evaluated solely on Single Domain Generalization, but still applicable here) in related work -- particularly since the broad components of the proposed approach in [A] and the current submission seem similar, including plausible O.O.D. instance generation & ensuring similarity across representations from augmented and non-augmented samples. This will further help highlighting the novelty and benefits of the proposed approach over [A].

- Since the proposed approach does not rely on domain labels, single domain generalization benchmarks are somewhat best suited to assess the utility of the proposed approach against other existing single DG approaches. While the authors conduct single DG experiments on PACS and DomainNet (more exhaustive on PACS, as highlighted in the supplementary) and compare with M-ADA, the paper would benefit from including comparisons with [A,B] (which outperform M-ADA) either on PACS / DomainNet or on the benchmarks used [A,B] -- CIFAR-10C or DIGITS or the semantic segmentation benchmarks.

- [Minor Points] -- It’s unclear what L63-64 is trying to convey in the introduction, revising the same would improve the paper. It’s unclear what L228-229 is trying to convey. Not a major point, but I think the opening line of the abstract should be repurposed to replace “task” with “models trained for a task”, it seemed confusing otherwise. The current draft would benefit from a little proofreading to address such minor issues.


[A] - Progressive Domain Expansion Network for Single Domain Generalization

[B] - Uncertainty-guided Model Generalization to Unseen Domains

**Post-Rebuttal Update**
My primary concerns about the paper (centered mostly around clarity and experimental comparisons) have been satisfactorily addressed. I also appreciate the detailed responses to the concerns raised by other reviewers. Given this, I am increasing my rating of the paper to 7.

**Time Spent Reviewing:**

7

---

> ### Author Response · Authors · 2021-08-10
> **Response to Reviewer 5PA5**
>
> **1. How exactly margin m is computed for the multi-source case (i.e., “calculated by the L2 distance between the means of the sampled source domains” in L178-181)? Is it the average L2 distance b/w the means across all possible pairs of source domains in the mini-batch? And how is the margin m selected for the single-source case? Is it chosen based on the target domain performance?**
>
> We thank the reviewer for giving us the opportunity to clarify the selection of margin m. For multi-source DG, we first calculate the means/centroid of data from each source domain in a mini-match, followed by averaging the L2 distances between the above centroid pairs. Since the largest number of source domains (for training) is 3 in the PACS dataset, calculation of the margin m would not be computationally demanding (i.e., 3 possible domain pairs). As noted in L179-181 and pointed out by the reviewer, this margin would be an important component, reflecting the expected feature discrepancy between data domains.
>
> Since the above learning strategy cannot be applied to single-source DG, we simply treat the margin m as a hyperparameter. We did not choose m based on the target domain performance. In fact, we simply fixed m = 0.1 throughout our experiments on PACS and DomainNet in Table 6, since we chose photo/real domain data as the source domain and always resize the images into 224 x 224 pixels as the inputs.
>
> To further assess the sensitivity of m in the single-source DG case, we additionally report the performances with different choices of m. For PACS using ResNet-50 as the backbone, we consider m as 0.001, 0.01, 0.1, and 1, and observe the average accuracies as 45.3%, 45.8%, 46.0%, and 44.0%, respectively. It can be seen that such performances are not dramatically sensitive to the m choices, while such results are consistently above those reported by SOTAs (e.g., Table 6).
>
> **2. How is the novel-domain augmentor realized? Did the authors experiment with other generators as choices for the augmentor design?**
>
> We thank the reviewer for the constructive remarks, and we are glad to provide more detailed discussions for our augmenter design. Following a recent augmentation-based DG work of [7], our augmenter is also realized by fully convolutional networks (FCN), composed of 1 conv. layer followed by 3 residual blocks and a 1 × 1 conv. layer. No pretraining is required for our augmenter. We will be happy to include this implementation detail into L217.
>
> We note that, such augmentation modules can be possibly implemented by different network architectures. As suggested by the reviewer, we additionally considered an AdaIn-based (i.e., translation-based) augmenter proposed in [A]. We observed a comparable accuracy of 84.4% (vs. 85.3%) on PACS, and a slightly degraded yet satisfactory accuracy of 65.5% (vs. 66.7%) on Office-Home. This suggests that, while we do not limit the design of our augmenter, the use of translation-based would be more preferable for the case in which the domain difference mainly exhibits visual style variations (e.g., PACS).
>
>
> **3. It will be helpful to highlight the novelty and benefits of the proposed approach over CVPR’21 single-source DG work [A].**
>
> We thank the reviewer for pointing out a relevant work recently published at CVPR 2021 [A]. Sharing the same goal of learning data augmentation for DG, [A] utilizes a progressive learning strategy, which iteratively expands the training data set by adding augmented data, while similar contrastive and adversarial learning objectives are observed. However, there are two potential issues/limitations for [A] as we discuss below.
>
> Firstly, [A] progressively learns to augment data, which will be recursively added into the training data set across different epochs (e.g., training data set size would be 10x larger than that of the initial one at the 10th epoch). Thus, it would be quite memory demanding for training [A]. On the contrary, our approach does not need to accumulate the augmented data, which does not produce the above concern.
>
> Secondly, [A] employs 4 losses (i.e., cycle consistency & adversarial & diverse & classification losses) in their proposed framework, which inevitably leads extra efforts on model tuning and hyperparameters selection. As for our proposed framework, only feature discrepancy and classification losses are needed. For multi-source DG, the margin m for the former loss (i.e., feature discrepancy loss) is determined by the source-domain data directly. As for single-source DG, we find that the choice of m would not dramatically affect the results, as we responded earlier.
>
> We will be happy to include the above discussion into Section 2 in our revised version.
>
>
>
> **4. Additional experiments with two CVPR 2021 single-source DG methods [A, B].**
>
> We are happy to carry out additional experiments of single-source domain generalization as suggested. Following two very recent CVPR’21 SOTAs of [A, B] using MNIST as the single source domain, our method showed promising results 61.7%, 83.2%, 69.3%, and 87.4% on SVHN, MNIST-M, SYN, and USPS, respectively. We achieved an average accuracy of 75.4% and performed against [A] and [B] which resulted in 74.8% and 61.3%, respectively. And, as discussed earlier, [A] has a much larger memory requirement and a more complex model (i.e., more hyperparameters to select). The above additional experiments further confirm the effectiveness of our method for single-source domain generalization.
>
>
> **5. Please clarify L63-64 and L228-229.**
>
> We thank the reviewer for giving us the opportunity to improve our paper. For L62-64, we now revise it as follows.
> “The objective is to maximize the discrepancy between the input and augmented data, derived from the teacher and student modules, respectively. In order to have such augmented data exhibit sufficient domain differences, the above discrepancy will be calculated using features derived from data across different source domains.”
>
> As for L228-229, we revise it below:
> “DeepAll is viewed as a baseline, in which both feature extractor and classifier are trained on data aggregated from all source domains.”
>
> We sincerely appreciate the comments and suggestions from the reviewer, and we will make proper adjustments or revisions in our future/final version.
>
>
> \
> [A] - Progressive Domain Expansion Network for Single Domain Generalization. CVPR 2021
>
> [B] - Uncertainty-guided Model Generalization to Unseen Domains. CVPR 2021

---

> > ### Comment · Reviewer_5PA5 · 2021-08-25
> > **Thanks for the response**
> >
> > Apologies for the delay. Thanks to the authors for providing a detailed response. Also, thanks to the other reviewers for pointing out bits that I had missed in my initial assessment of the paper — particularly regarding the choice to augment images rather than features (reviewer kqaD), errors in the max-margin formulation, justification of the generator objective, and other clarifications/errors (reviewer oj4V).
> >
> > My primary concerns revolved around 4 major points — clarification regarding the choice of the margin for multi-source and single-source experiments, clarity issues and justification regarding the choice of the generators, comparisons with relevant recent work [A, B] (experimentally and in terms of offered benefits and novelty). All of my concerns were addressed satisfactorily in the (strong) response provided by the authors. It’s great to see that the proposed approach is relatively robust to the choice of “m” for the single-source setting, the choice of the generator architecture, and is competitive with recent relevant work [A]. I also appreciate the experimental and clarification responses provided to the points raised reviewer oj4V.
> >
> > I think most concerns associated with the paper (including mine and other reviewers) seem to have been addressed. Given this, I am increasing my rating of the paper to 7.

---

### Official Review · Reviewer_GTRz · 2021-07-20

**Rating:** 7
**Confidence:** 3

**Summary:**

 This paper proposed adversarial teacher-student knowledge distillation for domain generalization. Motivated by recent progress in self-supervised representation learning, the authors proposed two learning stages including domain generalization representation learning and novel domain augmentation. During domain generalized representation learning, the student is updated while the augmenter is fixed and the loss minimizes the discrepancy between the teacher and student. During the Novel domain augmentation learning, the student is fixed and the augmenter is updated. The objective is to maximize the discrepancy and encourage the augmenter to generate more distinct samples. Experiments on several public benchmark datasets very the effectiveness of the proposed method.

**Ethical Concerns:**

N.A.

**Limitations And Societal Impact:**

N.A.

**Main Review:**

[Strength]

The paper is well written and the idea is novel. The authors perform extensive experiments and ablation studies to verify the effectiveness of each model.  The proposed 2 stage representation and augmentation learning dynamically combine cycle GAN and self-supervised learning by knowledge distillation. The insight that the teacher module is avoided from observing the unrealistic domain augmentation is interesting and verified from experiment results. The results are very good and achieve state-of-the-art performance on multiple benchmarks.

[Weakness]

In the Siamese archi. (From table 5), the paper shows optimizing both teacher and student through gradient descents still achieve reasonable performance. In self-supervised image representation learning, there will be severe mode collapses without EMA update. I wonder why siamese architecture is working here? Could the author illustrate more.

**Time Spent Reviewing:**

5 hours

---

> ### Author Response · Authors · 2021-08-10
> **Response to Reviewer GTRz**
>
> **1. In the Siamese archi. (From table 5), the paper shows optimizing both teacher and student through gradient descent still achieves reasonable performance. In self-supervised image representation learning, there will be severe mode collapses without EMA update. I wonder why siamese architecture is working here?**
>
>
>
> We thank the reviewer for pointing out the potential issue of mode collapse, which has been observed in recent self-supervised learning works like MoCo [16] and BYOL [17] while applying Siamese architectures. This is mainly due to the fact that such modules tend to result in a “shortcut” that outputs the same vectors regardless of different inputs to solve the contrastive pre-training task.
>
> As indicated by the reviewer, while our proposed framework achieved the best domain generalized classification results in Table 5, its simplified version (i.e., use of Siamese architecture) still reported satisfactory performance. We note that, it is because of the supervision of class labels during training of both versions of our model (i.e., Siamese archi. and Ours in Table 5). Nevertheless, the full version of ours (i.e., joint training of teacher-student modules with adversarial learning schemes) achieved the best results and thus would be preferable.
>
> While the SSL methods utilizing Siamese architectures can also be trained in a supervised manner (e.g., [A]) with alleviated mode collapse problems, they cannot be easily applied to DG tasks as ours does. This is mainly due to the fact that such models simply augment data via hand-crafted image transformations, which might not represent the domain shift.
>
> [A] Supervised Contrastive Learning. NeurIPS 2020

---

### Decision · Program_Chairs · 2021-09-27

**Decision:**

Accept (Spotlight)

**Comment:**

This paper was recommended for acceptance by all reviewers. The reviewers appreciated the fact that the paper was well-written, had comprehensive experiments, and presented a simple yet intuitive method. Though the reviewers originally had a few questions, the authors' response sufficiently answered those questions and convinced the reviewers of the merit of this work. The paper addresses the problem of domain generalization and the key aspect the reviewers appreciated the most was the relaxation of the assumption of domain labels and the effective performance shown from the method.